# Implementation of group interpersonal therapy to treat depression in people living with HIV: A first evaluation of IPT dissemination in Senegal

depression; HIV; sub-Saharan Africa; group interpersonal therapy; access to care

**Corresponding author:**
Charlotte Bernard;
Email: charlotte.bernard@u-bordeaux.fr

Hawa Abou Lam[1,2,3], Hélène Font[1,2], Véronique Petit[4], Salaheddine Ziadeh[5,6], Judicaël Malick Tine[7], Ibrahima Ndiaye[8], Ndeye Fatou Ngom[9,10], Babacar Ndiaye[11], Daniel Sarr[12], Dominique Diouf[12], Nathalie de Rekeneire[13], Antoine Jaquet[1,2], Moussa Seydi[7], Charlotte Bernard[1,2] 🆔 and the IeDEA West Africa Cohort Collaboration[1,2]

[1]National Institute for Health and Medical Research (INSERM), University of Bordeaux, Bordeaux, France; [2]Research Institute for Sustainable Development (IRD) EMR, Bordeaux, France; [3]Laboratoire de sociologie, anthropologie et psychologie sociale (LASAP), ETHOS, Université Cheikh Anta Diop, Dakar, Senegal; [4]UMR 196 CEPED (Centre Population et Développement), Université Paris Cité-Institut de Recherche pour le Développement (IRD), Paris, France; [5]Global Mental Health Lab, Teachers College, Columbia University, New York, NY, USA; [6]Faculté de Santé Publique, Université Libanaise, Sidon, Lebanon; [7]Service des Maladies Infectieuses et Tropicales, CHNU de Fann, Dakar, Senegal; [8]Service de Psychiatrie, CHNU de Fann, Dakar, Senegal; [9]Centre de Traitement Ambulatoire, CHNU de Fann, Dakar, Senegal; [10]Département de Médecine de l'Université Alioune Diop de Bambey, Bambey, Senegal; [11]Hôpital Youssou Mbargane Diop, Rufisque, Senegal; [12]Centre de promotion cardinal de la santé Hyacinthe Thiandoum, Dakar, Senegal and [13]Institut Pasteur du Cambodge, Phnom Penh, Cambodia

## Abstract

Group interpersonal therapy (IPT) was introduced to Senegal to treat depression in people living with HIV (PLWH), using a task-shifting approach. Following successful implementation at a tertiary-level hospital in Dakar, we evaluate IPT's acceptability, feasibility and benefits in primary and secondary-level suburban health facilities. We assess the impact of IPT adaptations and organizational changes and identify sustainability requirements. PLWH with depression received group IPT following the World Health Organization protocol. Acceptability, feasibility and implementation aspects were assessed quantitatively and qualitatively following specific conceptual frameworks. Depressive symptoms severity (PHQ-9) and functioning (WHODAS) were measured pre-, post-treatment and at 3-month follow-up. General linear mixed models were used to describe changes in outcomes over time. Qualitative data were analyzed thematically. Of 84 participants (median age: 45, female>50%), 81 completed group IPT. Enrolment refusal and dropout rates were 7% and 4%. Ninety-seven percent attended at least seven sessions out of eight. Depressive symptoms and functioning significantly improved by therapy's end ($\beta$ = 12,2, CI 95% [11.6, 12.8] and $\beta$ = 8.5, CI 95% [7.3, 9.7], respectively) with gains being sustained 3 months later ($p$ = 0.94 and 0.99, respectively). Adaptations and organizational changes proved successful, but depression screening and diagnosis communication to patients remained challenging. Emerging needs included a tailored patient care pathway and confidentiality. Participants advocated for depression care integration into HIV services. Group IPT's successful implementation in various ecological and organizational contexts in Senegal indicates high acceptability and feasibility. Sustainability may be enhanced by addressing specific needs at multiple levels (individual, organizational, systemic). A comprehensive reflection on strategies to sustain and scale up group IPT is the next logical step.





## Impact statement

This study evaluates whether group interpersonal therapy (IPT), recommended by the World Health Organization (WHO) to treat depression in resource-limited settings, can be effectively extended from centralized and privileged healthcare settings in a capital city to less centralized ones, thus assessing dissemination potential and identifying what could aid implementation. In addition to proven intervention applicability to different settings, the success of two implementation adaptations (*i.e.*, train-the-trainer model; opposite-sex group facilitation) shows capacity building sustainability and efficiency in using personnel resources.

This study was conducted in people living with HIV (PLWH), a vulnerable population in which depression is highly prevalent but remains underdiagnosed in Sub-Saharan Africa. We worked in Senegal, a West African country, where depression management in PLWH remains limited.

Our results confirmed high acceptability, feasibility and benefits of group IPT in various contexts in Senegal. Overall, adaptations made during this "decentralized" phase (*i.e.*, opposite-sex group facilitation; train-the-trainer model) were successful. These results underscore the flexibility and adaptability of group IPT, both important criteria in contexts with limited resources.

Our study informs future implementation and dissemination efforts. It identifies requirements to optimize depression screening and formally integrate depression care into HIV clinics. It confirms group IPT—combined with a task-shifting approach—as a promising intervention that could close the mental health treatment gap in Senegal. Further research is recommended to evaluate the long-term impact of group IPT and the potential for expansion across West Africa.

## Introduction

With 280 million people affected worldwide, depression is one of the most common mental health disorders and the leading cause of disability-adjusted life years globally (Vos et al., 2017). In low- and middle-income countries (LMICs), up to 79%–93% of individuals with depression do not get access to treatment (Chisholm et al., 2016). Lack of mental health specialists, insufficient resources dedicated to mental health care and poor integration of mental health services can explain this unmet need for depression treatment (Lancet Global Mental Health Group et al., 2007; World Health Organisation, 2010; Patel et al., 2011). The World Health Organization (WHO) recommends psychotherapy as a first-line treatment, particularly group interpersonal therapy (IPT) in LMICs, as well as task-shifting (Lancet Global Mental Health Group et al., 2007; WHO, 2008), widely recognized as an effective method to implement psychological interventions (Patel and Thornicroft, 2009; Patel et al., 2010).

Group IPT is a time-limited psychotherapy that consists of eight weekly group sessions and a single pre-group individual session (WHO, 2016). A key premise of IPT is that interpersonal problems can trigger depressive symptoms. In sessions, patients learn and develop skills to better manage their interpersonal problems (Weissman et al., 2018). IPT has been successfully adapted to different conditions, intervention modalities, age groups and care settings worldwide, particularly in other African countries (Mootz and Weissman, 2024; Weissman and Mootz, 2024). Its effectiveness in treating depression has been demonstrated in systematic reviews, though some mitigated results have been observed – as it was the case with other tested psychotherapies (Cuijpers et al., 2011, 2016; Cohen et al., 2024). Group IPT was first applied in sub-Saharan Africa (SSA) with positive results in Uganda (Bolton et al., 2003), and shown to be both effective and well-suited when combined with a task-shifting approach (Lancet Global Mental Health Group et al., 2007; World Health Organisation, 2010).

Group IPT has been applied to treat depression in people living with HIV (PLWH) in SSA (Petersen et al., 2014; Asrat et al., 2020, 2021; Meffert et al., 2021). Since depression is the most common neuropsychiatric disorder among PLWH (Abas et al., 2014) and is associated with negative HIV outcomes (*i.e.*, poor adherence to antiretroviral therapy (ART), faster progression to AIDS, slower CD4 recovery) (Berhe et al., 2013; Memiah et al., 2014; Wroe et al., 2015) and reduced quality of life, integrating depression treatment into HIV care becomes critical (Abas et al., 2014).

Our study, the first to implement group IPT with a task-shifting approach (*i.e.*, conducted by social and community health workers) to treat depression in PLWH in Senegal, was multi-phasic. The first phase (hereafter, "*PHASE-1*") ran from March 2019 to March 2022, at the Fann National University Hospital Center (FNUHC) (level 3 hospital), in Dakar, the capital and largest city in the country, and was carried by the Infectious and Tropical Diseases Unit (SMIT) and the Outpatient Treatment Center (CTA) at the FNUHC. High acceptability, feasibility and benefits for 60 PLWH were observed, including significant improvements of depressive symptoms with sustained gains 3 months after the end of therapy, in addition to improvements in PLWH's social and professional lives, and enhanced clinical competence for professionals (Bernard et al., 2023, 2024). Conducted in a tertiary-level hospital in the capital, this successful *PHASE-1* reflected a rather centralized system of care. Thus, the results could not be generalized to less centralized settings, such as peripheral and rural areas of LMICs, where significant gaps in mental health care prevailed (Parcesepe et al., 2018). Expanding group IPT beyond Dakar to these areas could help reduce the disparity gap.

Building on *PHASE-1* success, we implemented group IPT in primary and secondary-level health care facilities outside Dakar (*i.e.*, the Youssou Mbargane Diop Hospital and the Hyacinthe Thiandoum Cardinal Health Promotion Center), from March 2022 to August 2024 (hereafter "*PHASE-2*"). Using a mixed-methods approach, the *PHASE-2* study aimed to: (1) confirm group IPT acceptability, feasibility and benefits in these different contexts; (2) assess the impact of group IPT adaptations and organizational changes between *PHASES 1 and 2*; and (3) determine requirements for sustainability.

## Methods

### Study design

The study was conducted as part of a research agenda through the International Epidemiological Databases to Evaluate AIDS (IeDEA) West Africa collaboration (http://iedea-wa.org/). Both quantitative and qualitative methods were used to describe acceptability, feasibility and benefits of group IPT for patients and facilitators, with pre-/post-intervention evaluations. In the quantitative descriptive part of the study, we compared depressive symptoms and disability in PLWH treated for depression with group IPT pre-intervention, post-intervention and at 3-month follow-up.

### Group IPT

#### Delivery

Group IPT was delivered according to WHO manual guidelines (World Health Organization, 2016), translated to French by our team (https://www.who.int/publications/i/item/WHO-MSD-MER-16.4). It was delivered over the course of eight weekly group sessions, preceded by one pre-group individual meeting. The eight sessions covered the three group IPT phases: initial, middle and termination. Therapy was provided in French and/or in Wolof, Senegal's national language. Each group consisted of 6 patients and one group facilitator. Facilitators were social workers and community health workers. In Senegal, a social worker typically holds a state diploma in social work, obtained after 3 years of training following the high school diploma. Community health workers are generally recruited and trained within the facilities providing support to PWH. In addition

to their experience of the disease, they are trained in essential health concepts, communication techniques and practical tools for outreach work.

### Training

Training of group IPT facilitators was led by a clinical psychologist (hereafter "Master Trainer"). To qualify as group IPT facilitators, trainees had to: (1) attend a 5-day training workshop on IPT principles, strategies and techniques; and (2) facilitate 3 groups: first as co-facilitators, then as solo facilitators for the subsequent two.

Facilitators received an additional 4-h depression training with a local psychiatrist at their corresponding site. Once groups were formed, facilitators began weekly online supervision sessions (90 min each) with the Master Trainer.

### Adaptations to local context

Group IPT was adapted to the local context, presented in detail elsewhere (Ziadeh et al., 2024). In *PHASE-2*, other adaptations were needed (Figure 1). First, the Master Trainer conducted the 5-day training remotely (due to COVID travel restrictions). Second, the training incorporated clinical examples from *PHASE-1*. Third, with no male social workers at one facility – a common situation in Senegal, some groups receiving IPT were conducted by facilitators of the opposite sex of patients. Though inconsistent with Senegalese norms and previous practice, this adaptation was deemed necessary. Perceptions of all aforementioned adaptations were documented.

### Main organizational changes between PHASES 1&2

The main organizational changes between *PHASES 1&2* are presented in Figure 1. First, in *PHASE-2*, depression screening was conducted by either a social worker or a community health worker, with subsequent diagnosis confirmation by the referring doctor, not a mental health specialist. In contrast, during *PHASE-1*, screening and diagnosis were performed by the referring doctor and a psychiatrist, respectively. Second, facilitators from *PHASE-1* were trained to become group IPT supervisors. This involved: (1) attending a 4-h training on supervision principles; and (2) participating in supervision sessions with the Master Trainer as co-supervisors of the new group facilitators, gradually proceeding to independent supervision. In *PHASES 1&2*, seven were trained as group facilitators (four in *PHASE-2*) and three were trained as supervisors. This train-the-trainer approach in mental health care was new in Senegal, especially with community health workers supervising social workers. Perceptions of those changes were explored.

### Study population

Depression screening was offered to PLWH during their routine HIV clinical visits, using the Patient Health Questionnaire (PHQ-9) (Kroenke et al., 2001). To be included, PLWH had to be adults aged ≥20 years, on ART, and have a confirmed diagnosis of depression. PLWH with a PHQ-9 score ≥ 10 (standard cut-off) were referred to the attending physician, who confirmed or rejected the diagnosis using the MhGAP depression module (World Health Organisation, 2010). Exclusion criteria included: hospitalization and/or medical emergencies; diagnosis with a psychiatric illness other than depression; vision or hearing impairment that seriously affects group interactions; and imminent suicide risk (in which case, the candidate was immediately referred to the psychiatrist).

For the qualitative interviews, we interviewed 22 PLWH who had completed group IPT. To include a variety of perspectives, the sample was selected using maximum variation sampling (Green and Thorogood, 2018) with regard to age, marital status, profession and levels of suicidal risk. Selection also depended on participant availability. To gather insights from various stakeholders involved in the project, we conducted interviews with each facilitator and supervisor at two time points: the end of training (6 interviews) and the end of the study phase (6 interviews). Additionally, we interviewed physicians and heads of services based on their availability (3 interviews).

### Quantitative data collection and outcomes

Acceptability was measured through the refusal rate (the number of patients with depression who declined to participate in group IPT divided by number invited), and the drop-out rate (the number of patients who started the group therapy but did not continue attending the sessions divided by the number of those who agreed to participate) (Purcell et al., 2007; Craig et al., 2008; Patel et al., 2011; Proctor et al., 2011). Patient satisfaction with this

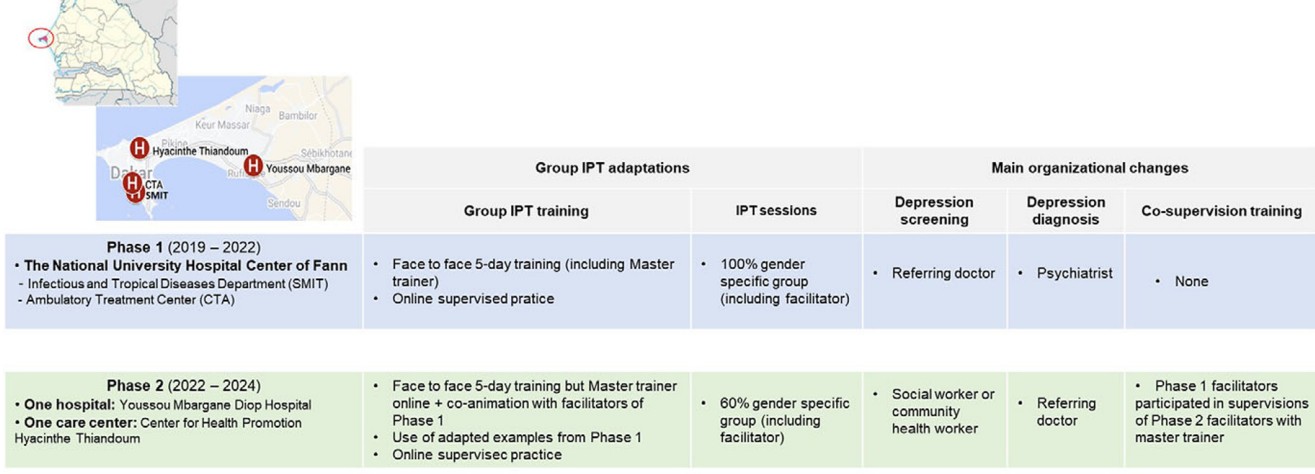

**Figure 1.** Settings, group IPT adaptations and main reorganizations between the two *PHASES* of the project.

therapy was evaluated using two questions adapted from the Consumer Satisfaction Questionnaire (CSQ-8) (Kapp et al., 2014), evaluating the quality of the service they received and their level of satisfaction, using a 4-point Likert scale. Feasibility was assessed using various indicators (Bowen et al., 2009; Proctor et al., 2011) including attendance of therapy sessions, risk of suicide during group IPT delivery, duration of treatment (*i.e.*, 8 weeks or more), and time between confirmation and the start of group IPT. We also asked facilitators and PLWH whether they would recommend this therapy to others.

Clinicians assessed depressive symptoms and disability at baseline and during follow-up visits (*i.e.*, at the end of group IPT and 3 months later), using the PHQ-9 and the World Health Organization Disability Assessment Schedule (WHODAS-12) (Üstün et al., 2010), respectively. The PHQ-9 was used to evaluate the severity of depressive symptoms. Each item was rated on a 4-point Likert scale ranging from 0 to 3. A total score on the PHQ-9 was obtained by summing up the scores on the individual items and could range from 0 to 27. The total score indicated the severity of depressive symptoms: mild (5–9), moderate (10–14), moderately severe (15–19) or severe (≥20). The PHQ-9 is widely used in studies conducted with PLWH in SSA (Wagner et al., 2011; Belenky et al., 2014; Endeshaw et al., 2014; Musisi et al., 2014; Asangbeh et al., 2016) and is recommended for use by the mental health work group of the IeDEA Cohort collaboration (NIMH).

Suicide risk was systematically assessed for each patient who reported anything other than "not at all" for the last item of the PHQ-9 (*i.e.*, "In the last 2 weeks, have you thought about hurting yourself in any way?"). We used the Columbia Suicide Severity rating scale, a standardized tool to assess the level of risk (passive, active-low, active-moderate, active-severe) (Posner et al., 2011). Patients with an active moderate-to-severe risk were immediately referred to the psychiatrist for care.

The WHODAS 12-item measure uses a 5-point Likert scale (*i.e.*, no difficulty, mild difficulty, moderate difficulty, severe difficulty, extreme difficulty/"cannot do") to assess functioning across 6 domains (*i.e.*, cognition, mobility, self-care, getting along, life activities and participation in community activities) (Üstün et al., 2010). A total score is obtained by summing up item scores and could range from 12 to 60, with higher scores denoting more loss of function. The WHODAS has shown good psychometric properties and was used in several countries (Federici et al., 2017), particularly with people living with mental illness or chronic disease in Ethiopia (Habtamu et al., 2016), Nigeria (Igwesi-Chidobe et al., 2020) and Ghana (Badu et al., 2021).

### Qualitative data collection and outcomes

Semi-structured individual interviews were conducted in French (Senegal's official language and Wolof (Senegal's national language) by a socio-anthropologist (HAL) on site from February 2023 to June 2024. The Consolidated Framework for Implementation Research (CFIR) (Damschroder et al., 2009), Theoretical Framework of Acceptability (TFA) (Sekhon et al., 2017) and the conceptual framework on feasibility (Bowen et al., 2009) were used to develop the interview guides and data analysis. These frameworks offered a comprehensive approach to explore the intervention from multiple dimensions. Additional themes related to perceptions and experiences were added to meet the research objectives. The interview guides were refined mid-term based on field observations and discussions with the research team. The interviews (30–120 min each) were audio-recorded and then transcribed by the interviewer. Interviews conducted in Wolof were

contemporaneously translated and transcribed in French; those in French were transcribed *verbatim.* Following transcription, all recordings were deleted and interview transcripts were anonymized. Excerpts included in this manuscript were translated from French into English. This study part followed the guidelines of the Standards for Reporting Qualitative Research (O'Brien et al., 2014; Thébaud and Dargentas, 2023).

### Data analysis

#### Quantitative data

The characteristics of the participants, and acceptability and feasibility indicators, were described using median and interquartile range (IQR) for continuous variables and counts and proportions for categorical variables. Significant improvement of depressive symptoms and disability scores were assessed between (1) inclusion and the end of group IPT and (2) between the end of group IPT and 3 months post-treatment, using a general linear mixed model with random intercept including time as fixed effect and patient as random effect ($p < 0.05$). Observations with missing data were excluded (<10). A description of the different scores is presented with mean and standard deviation for each visit. The evolution of the PHQ-9 symptoms is presented in an alluvial plot (Rosanbalm, 2015), according to symptom severity. The impact of opposite-sex facilitated groups on the evolution of depressive symptoms was evaluated using a general linear mixed model with random intercept including time, opposite-sex facilitation (yes/no-binary variable), and their interaction as fixed effects and patient as a random effect.

#### Qualitative data

Interviews were analyzed using a classical thematic analysis. Themes were identified following a deductive approach, based on the selected theoretical frameworks with a focus on changes between the end of training and the end of the study phase, as well as group IPT sustainability. A coding tree was developed based on these themes by HAL (socio-anthropologist), then discussed and reviewed by VP and CB, with expertise in qualitative approach and mental health. Strategies to enhance rigor included focused observations and discussions of emerging themes with the treatment team, along with cultural contextualization. All facilitators and supervisors were interviewed. For PLWH, data analysis continued until thematic saturation was reached. All transcripts were read and coded using NVivo version 1.7.1. Quantitative and qualitative data were first analyzed separately, then merged and organized by themes.

### Ethical approval.

The research was conducted in accordance with the Helsinki Declaration. Ethical approval was obtained from Senegal's ethics committee: Conseil National d'Ethique de la Recherche en Santé (CNERS) (Protocol-number: SEN22/49; approved-number: 0000102/MSAS/CNERS/SP). Before taking part in the study, all participants gave written consent, collected in a private room reserved for medical visits. The collected data were anonymized.

## Results

### Study participants

Two hundred and fifty-four patients were screened for depression. Ninety-four got a confirmed diagnosis of depression. Seven refused to participate in therapy. One patient died before starting therapy

**Table 1.** Characteristics of the sample

| Variables | *N* (%) |
|---|---|
| *N*[a] | 84 |
| Age (years) (median IQR) | 45 (38–53) (Min: 20/Max: 70) |
| Female | 53 (63.1) |
| Living alone | 47 (56.0) |
| Unemployed | 26 (31.0) |
| HIV status not shared | 33 (39.3) (mis.: 2) |
| Financial difficulties | 75 (89.3) (mis.: 1) |
| HIV infection duration (years) (median IQR) (mis.: 3) | 11 (6–13) (Min.: 0.1/Max: 19) |
| High antiretroviral treatment adherence at baseline[b] | 83 (98.8) |

Abbreviations: IQR, interval interquartile; mis., missing data.
[a]Including patients who dropped out group IPT.
[b]Adherence to ART was evaluated as the percentage of tablets the patient declared to have taken over 7 days (in comparison to the prescribed total number of tablets over this period). High adherence is considered when the percentage is 100%.

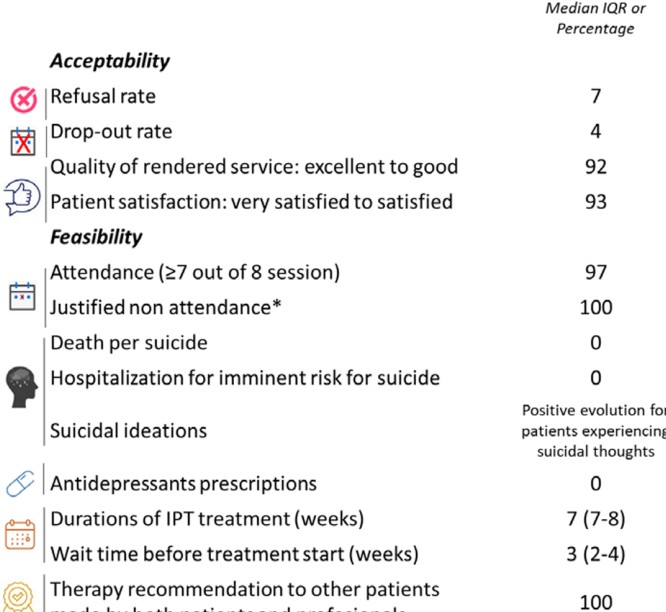

**Figure 2.** Main acceptability and feasibility outcomes.

(general medicine). One accepted but did not participate in any session (end of protocol). Data from one patient were excluded from the analysis due to incorrect completion of their consent to participate. Eighty-four PLWH with depression were included. Median age was 45 years old, >50% were female, 31% were unemployed (Table 1).

In total, 14 IPT groups were formed. In the course of treatment, three patients dropped out. In contrast, 81 patients completed their group IPT treatment.

PLWH attributed their depression mainly to life changes (44.6%) and conflicts (31.3%). Grief and social isolation were less reported (13.3% and 11.1%). Among the patients who completed group IPT, more than 90% attended their follow-up visit at the end of IPT. Eighty-nine percent came to their 3-month post-treatment follow-up.

### Main acceptability and feasibility outcomes

The refusal rate was low (7%) (Figure 2). The drop-out rate was 4%. Dropout occurred after attending the pre-group session (two patients) or at least two group sessions (one patient).

The quality of rendered service was rated as "excellent" or "good" by 92% of the participants, and patient satisfaction was high (93%). All participants agreed that group IPT was suitable and beneficial for PLWH with depression. PLWH reported positive experiences with group IPT, where they formed friendships, gained new support, improved their self-esteem and learned to cope with depression.

> Therapy has allowed us to free ourselves, to talk about our suffering and to find a solution for some of our problems, and even helping a person find a solution to their problem, it makes you happy. (PLWH 3)

The waiting time before treatment start was a median of 3 weeks. Attendance was high (97%), with 100% of nonattendance justified (*i.e.*, illness, death in the family, work constraints). A positive evolution of suicidal ideations was observed without the need for hospitalization. The duration of IPT treatment was respected despite occasional rescheduling of sessions. Both patients and

professionals unanimously recommended group IPT. Facilitators reported key prerequisites for success: confidentiality, therapist-participant trust and clear communication of therapy goals. They also raised schedule conflicts as a barrier, particularly for working participants. Verbatim excerpts illustrating this part are available in the Supplementary Data (Table S1).

### Depressive symptoms reduction and other benefits of group IPT

At baseline, 74% had moderate symptoms and 26% had moderately severe to severe symptoms (Figure 3). By the end of treatment, 6% had mild depressive symptoms. Three months' post-treatment, 5% had mild depressive symptoms. Patients with mild depressive symptoms at the end of group IPT had no symptoms 3 months later ($N = 3$) or remained with mild depressive symptoms ($N = 1$). However, 3 patients with no depressive symptoms at the end of group IPT had mild symptoms, including sadness or anhedonia, 3 months later. The mean PHQ-9 score was 13.4 (SD = 2.6), 1.2 (SD = 1.9) and 1.2 (SD = 1.6), at baseline, end of group IPT and 3 months later, respectively. Symptoms improvement was statistically significant between baseline and the end of therapy ($\beta = 12,2$, Confidence Interval (CI) 95% [11.6, 12.8] $p < 0.001$) but no significant change was noted from the end of group IPT to the 3-month follow-up ($\beta = 0.02$, CI 95% [−0.6, 0.6] $p = 0.936$) (Supplementary Table S2, left). The mean WHODAS score was 21.5 (SD = 6.3), 13.0 (SD = 2.1) and 13.0 (SD = 2.1), at baseline, end of group IPT and 3 months later, respectively. A significant improvement in WHODAS scores from baseline to the end of therapy was observed ($\beta = 8.5$, CI 95% [7.3, 9.7], $p < 0.001$), but no significant change was noted from the end of group IPT to the 3-month follow-up ($\beta = 0.01$, CI 95% [−1.2, 1,2], $p = 0.99$) (Supplementary Table S2, right).

As above, depressive symptoms and disability reduction, PLWH also reported improvements in their daily, professional and social lives as well as better acceptance of HIV disease and a better

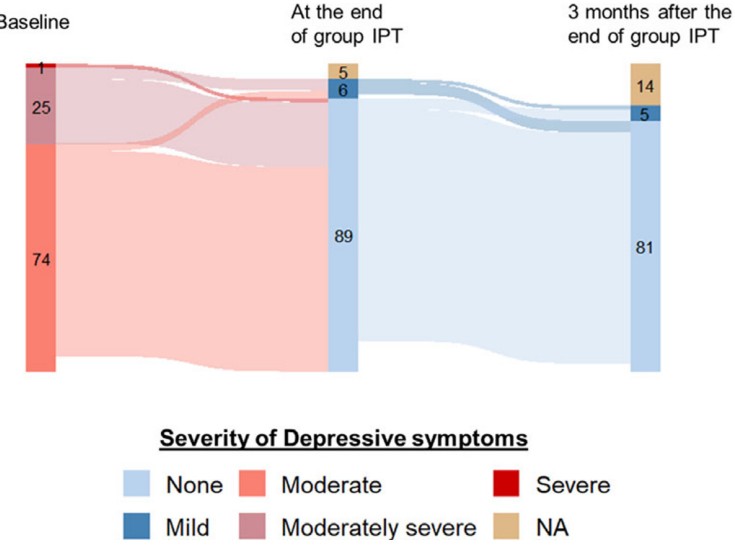

**Figure 3.** Description of the prevalence and the evolution of depressive symptoms according to severity, at baseline, end of group IPT and 3 months after the end of group IPT.

commitment to care. Facilitators and supervisors also reported the positive impact of group IPT on their professional abilities (better understanding of patients' difficulties, better knowledge about depression, improvement in their professional skills) (Verbatim excerpts Supplementary Table S1).

### *Impact of adaptations of group IPT*

#### *Impact of training adaptations*
The importance of adapting the training content to the local context was emphasized. Case examples from *PHASE-1* incorporated into the training were well appreciated as concrete local examples of group IPT applications. The Master Trainer conducted the training online, which was a new experience for trainees. One needed time to adjust to remote learning, particularly the physical absence of the trainer. Difficulties of concentration and understanding were reported, then resolved through practice and formative supervision.

*Well, the textbook is only (…) a guide (…) we have to take into account the sociological contexts [of cases], because they are different.* (Facilitator-2)

*I was used to doing training where I had direct contact with the trainer, (…). [In this training], we saw S. [Master Trainer] but in video conference, so we weren't so used to this method (…). In the beginning, it took me a lot of concentration to be able to understand and keep up (LAUGHTER).* (Facilitator-1)

#### *Impact of opposite-sex facilitated groups on depressive symptoms evolution and acceptability*
The gender of the facilitator, being different from PLHW included in group IPT, had no significant effect on overall PHQ-9 scores ($p = 0.35$) (Figure 4, Supplementary Table S3). No significant interaction was found between group type and score changes at either time point ($p = 0.98$ and $p = 0.64$, respectively). In addition to not affecting the evolution of symptoms, this change in the

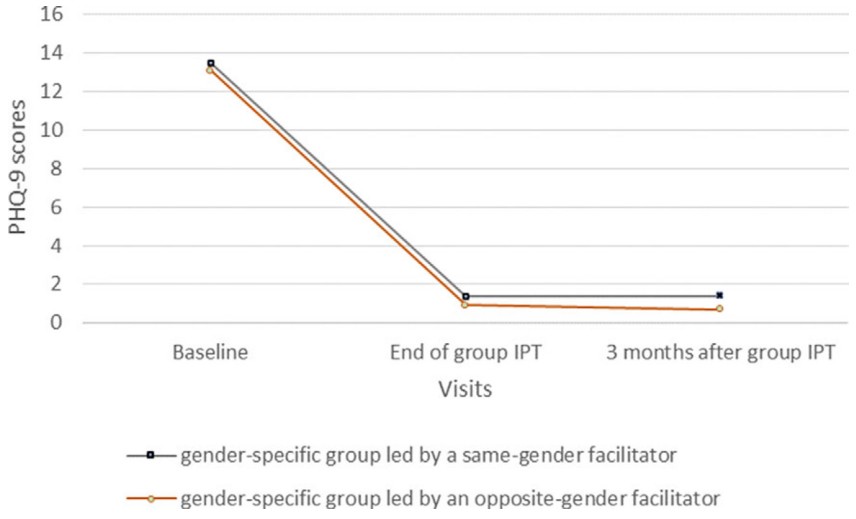

**Figure 4.** Evolution of PHQ-9 in function of facilitator gender, being different from PLHW included in group IPT.

organization was well received by PLWH and facilitators. For PLWH, the gender of the facilitator was not important as long as the job was done. Facilitators reported that they had been able to adapt successfully and identified trust between patient and therapist as a condition for successful adaptation. One female facilitator reported a preference for opposite-sex group facilitation, while a male facilitator said he felt more comfortable with females as they expressed themselves more easily than males.

*They [patients] look at us as health workers, as providers (…) The challenge is to do everything possible to establish a relationship of trust. (…) the rest is so easy (…) You have to know how to adapt approaches.* (Facilitator-1)

*As long as it's for the betterment of the person, as long as it takes away the person's suffering, I don't mind. The main thing is that the person is mature, has the abilities, whether he or she is a man or a woman.* (PLWH-2)

### Impact of the main organizational changes

#### Persistent problems at the screening step

Despite the decision in *PHASE-2* to delegate screening to social workers trained in group IPT, access to screening did not improve. Systematic screening was difficult to implement. In both sites, confusion of roles and responsibilities was observed. There appeared to be a lack of organizational coordination, which disrupted the order of the implementation process. For example, social workers often discussed group IPT participation with potential patients, even when a diagnosis of depression had not yet been established.

*Some patients came very early, so they went directly to either me or Dr. XX. So, that's why not all patients could go to the social service office.* (Physician-1)

*Even before doing the screening, we welcome the patient, we first explain the project, how it is goes and that it is a project, and that they came to help us, so we had to apply the form, the PHQ, to see if the patient can benefit from the therapy or not.* (Facilitator-1)

#### Impact of a less medicalized approach on treatment

Announcing a diagnosis of depression proved challenging to non-mental health specialists. Group facilitators were concerned that a direct use of the word "depression" could deter patients from attending group IPT, given the stigma and misconceptions associated with this term in Senegal.

*At the beginning we were a little cautious, we went little by little (…) and in the end we integrated the notion of depression (…) We didn't want to tell them at the very beginning that you have depression, […] I wasn't afraid that they would stop coming to see the attending physician for their medical follow-up, but it was more, that they would start to miss therapy sessions.* (Facilitator-2)

The initial avoidance of the term "depression" delayed patient understanding of their condition and treatment objectives in the first moments of *PHASE-2*. Some patients initially conflated group IPT with discussion groups they used to attend as part of their regular HIV care.

*I can't say it's a disease[depression]. If I take my case, when I talked about it, I felt relieved and it wasn't what I thought it was. But maybe if you hold on and don't talk, in the long run, it can become a disease.* (PLWH-1)

Gradually, with more experience conducting group IPT sessions, some facilitators were able to overcome their initial hesitation to

explicitly discuss depression during implementation, whereas for others, the issue persisted to some extent.

*Well, maybe we changed the discourse just after the second group, but we were still very careful. After the second group, I used the word depression directly.* (Facilitator-2)

#### Experience in a new supervisory role

Facilitators from *PHASE-1* who trained in supervision during *PHASE-2* and co-supervised the new cohort of facilitators described their new supervisory role as both interesting and rewarding. Despite initial doubts and apprehensions about their abilities, they ultimately felt comfortable in their role and were able to achieve their objectives.

*The training capitalized on us; it was really of good quality.* (Supervisor-1)

*We were supervised [in PHASE-1], but we never supervised anyone (…). I saw that it was a very interesting opportunity, (…) I supervised, I learned at the same time, it was really great (…). I saw that she [Supervised-facilitator] learned from me. I learned from her (…). Initially, I was scared maybe, but afterwards, I was comfortable. Any person you give me now I can supervise.* (Supervisor-1)

Facilitators in *PHASE-2* reported that their supervisors-in-training were well-prepared, providing valuable insights that enhanced their understanding of group IPT practice; supervision helped them feel supported and confident in their role as facilitators.

*The supervision was interesting (…) We did it in a very relaxed atmosphere and the supervisors were really up to the task.* (Facilitator-1)

*The supervisor (…) hads some experience. We learned from him. We know that he masters therapy. There are days, especially the last therapy sessions, where S. [Master Trainer] came every fortnight, and our [supervisor-in-training] was in charge of supervision the following week. (…) So we learned from them both.* (Facilitator-2)

### Sustainability and scale-up

#### Site sustainability needs

To sustain group IPT in routine practice, strongly advocated by participants, several needs were identified. At the individual level, PLWH and professionals reiterated the importance of reimbursing transportation costs, suggesting the amount should be proportional to the traveled distance. Further, some facilitators called for solutions tailored to the needs of working PLWH, such as offering them individual therapy.

*The question of transport can be reassessed according to distances that patients travel, since we have set a sum that does not do justice to [people from] different geographical areas. […] we may have to look at this in the future.* (Site-manager-1)

At the organizational level, the professionals involved in the project emphasized the importance of tailoring patient care pathways to each specific facility, of training and organizational adjustments to accommodate group IPT in their work routines. They also highlighted the need for staff commitment and making suitable rooms available for group therapy.

*This start by reviewing the patient's care pathway (…). This will allow us to systematically screen all patients who come to the hospital.* (Physician-1)

*I alone cannot initiate therapy […] without the commitment of other colleagues. Because while I am committed to conducting the therapy, others should take charge of the rest.* (Facilitator-1)

At the systemic level, professionals emphasized the critical need to integrate depression screening and management into the healthcare protocols of PLWH within ministerial directives. Without such integration, the sustainability of the intervention at sites will be challenging.

> *Obviously it's very feasible and acceptable. The main thing is for the authorities to carry the project afterwards and have the will (…) to support the project and disseminate it to other sites.* (Facilitator-2)

> *We want to wait for the authorities' response to see if the therapy gets integrated […] since it is a public service (…). We apply the protocols we use, which come to us from up there.* (Facilitator-1)

> *I think that if the mechanism is followed by a desire from our leaders to implement it, it could help the structure in the care for patients.* (Site-manager-1)

### Scaling-up feasibility

PLWH and professionals expressed strong support for the expansion of group IPT, despite several barriers. Some indicated that implementation in other hospitals or health centers could succeed if sessions could be organized with the care team within those facilities and if therapy objectives were clearly communicated to PLWH. Others highlighted challenges associated with healthcare services in remote areas, particularly large distances between villages and care facilities, potential language barriers and issues of confidentiality – everyone knows everyone in small villages.

> *In rural areas, it's feasible. But (…) If you don't speak the language, you can't communicate. The tools will have to be adapted, the facilitator will have to speak the local language, and perhaps the problem of accessibility: because there are people who live in distant localities.* (Facilitator-3)

## Discussion

This study confirmed high acceptability, feasibility and benefits of group IPT in different contexts in Senegal. Key findings included high attendance, low drop-out rates, significant symptom improvement and maintenance of gains 3 months post-intervention. Adaptations made between *PHASES 1&2* did not adversely affect success. Notably, opposite-sex facilitation was successful, as was the adoption of the train-the-trainer model. However, access to depression screening remained limited, and depression diagnosis communication is yet to be improved. To ensure sustainability, several implementation adjustments at various levels were identified: equitable transportation reimbursement, schedule flexibility with working patients, adequate delineation of patient care pathway and systemic integration of depression treatment into HIV services. Further, limited access to healthcare facilities, language barriers and confidentiality concerns were reported as major challenges to group IPT dissemination in rural areas.

Compared to *PHASE-1*, impacted by lockdown and fears related to visiting facilities due to the COVID-19 pandemic, *PHASE-2* produced equivalent or better results in terms of session attendance and patient retention (Bolton et al., 2003; Petersen et al., 2014; Meffert et al., 2021). In both *PHASES*, participants recognized group IPT as beneficial for treating depression. In *PHASE-2*, one specific adaptation was made to facilitate implementation: whereas we required same-sex group facilitation in *PHASE-1*, based on the premise that cultural or gender dynamics could influence interpersonal interactions (Nakimuli-Mpungu et al., 2014; Petersen et al.,

2014; Rose-Clarke et al., 2022). Thus, we experimented with opposite-sex facilitation and found the first evidence of its suitability, at least in Senegal. Associated findings, such as good outcomes for PLWH and positive experiences for facilitators, are encouraging in that it decreases the likelihood of opposite-sex patients being left without care.

Concerning organizational changes, a train-the-trainer strategy was used, whereby facilitators from *PHASE-1* participated in the training and supervision of the new facilitators in *PHASE-2*. Trainees reported a positive, collaborative learning environment, which is presumed to enhance effective training delivery (Esponda et al., 2020). This train-the-trainer model has been successfully used in several mental health programs worldwide, with limited resources and to facilitate dissemination (Madah-Amiri et al., 2016; Nakimuli-Mpungu et al., 2021; Lawson et al., 2023). It has been described as cost-effective, inasmuch as it decreases the need for external expertise over time (Raghavan et al., 2024). Our two successful adaptations highlighted the flexibility and adaptability of group IPT, both important criteria in contexts with limited resources.

Despite some adjustments made, systematic screening for depression remained difficult. Specific measures to integrate systematic screening in SSA countries have been recently reported. These include specific training, a clear patient pathway that depends on setting characteristics, and integration of depression care into HIV clinics (Guichard et al., Under review; Grimes et al., 2024). In addition to resource constraints, the other challenge that we encountered was mental health perception, which is consistent with reports in other SSA countries, where cultural beliefs and stigmatization are important barriers to treatment, a mental illness often associated with madness or witchcraft (Memiah et al., 2014; Martin et al., 2020; Molebatsi et al., 2022; Grimes et al., 2024). As observed, there were difficulties in naming depression as such or giving the patient the diagnosis of "depression" (especially for non-specialists). A primary concern for the clinical team was that such a diagnosis, with its connotations, might discourage potential patients from participating in treatment. Apprehensive, they may not be able to listen through to understand and comprehend what the treatment can do for them. In Ethiopia, where similar difficulties were observed, avoiding a direct mention of depression reduced anxiety and resistance among participants (Wondimagegn et al., 2024). However, it was also our observation that training and supervision provide critical steps in learning how to discuss depression with PLWH; they clarify professional roles and tasks throughout the entire process of care delivery (*i.e.*, avoid discussing treatment options at screening). Improving mental health literacy is also urgently needed to empower patients, to encourage help-seeking behaviors and facilitate screening and treatment.

Integrating depression care into HIV services requires policy changes to include depression management in national HIV treatment guidelines. This would provide the opportunity to identify strategies for dissemination and training funding sources, supervision and transportation costs, all of which are critical to enhance access to care (Asrat et al., 2020; Meffert et al., 2021). Adaptation of care provision emerges as necessary for employed PLWH, and those who live in rural communities, as well as key populations. While individual IPT might be a viable alternative, it should be implemented selectively, when necessary, in part not to overwhelm mental health staff with increasing time demands. Telephone-delivered therapy, which has proved its potential in SSA (Miniati et al., 2023; Mayberry, 2024; Meffert, 2024), would be another

option, especially in rural areas where it could circumvent the issues related to distance and confidentiality.

This study represents a pioneering initiative in West Africa whereby the treatment of depression in PLWH, based on a task-shifting approach, was disseminated to decentralized settings (*i.e.*, primary and secondary healthcare facilities outside the capital). Nonetheless, the study had some limitations. First, depressive symptoms after the end of therapy were only assessed at 3-month follow-up. The longer-term impact of group IPT on participants' depressive state was not evaluated. Second, participants, especially professionals, may have shared positive experiences with group IPT due to a social desirability bias. Such a bias, however, should be minimal since quantitative and qualitative results were consistent with one another.

## Conclusion

The study demonstrated that group IPT was both feasible and acceptable in different contexts of HIV services in Senegal, and that it had significant positive impacts on depressive symptoms evolution and beyond. The urgent need for a systemic integration of depression management into HIV care has become evident within the broader context of integrated healthcare. A comprehensive reflection on strategies to ensure the sustainability of group IPT in HIV care services and optimize its scale-up across the country is now necessary.

**Open peer review.** To view the open peer review materials for this article, please visit http://doi.org/10.1017/gmh.2025.10029.

**Supplementary material.** The supplementary material for this article can be found at http://doi.org/10.1017/gmh.2025.10029.

**Data availability statement.** The data that support the findings of our study are available from the corresponding author upon reasonable request.

**Acknowledgments.** The authors would like to thank the IeDEA West Africa region: Site investigators and cohorts:
Adult cohorts: Marcel Djimon Zannou, CNHU, Cotonou, Benin; Armel Poda, CHU Souro Sanou, Bobo Dioulasso, Burkina Faso; Fred Stephen Sarfo and Komfo Anokeye Teaching.
Hospital, Kumasi, Ghana; Eugene Messou, ACONDA CePReF, Abidjan, Cote d'Ivoire; Henri Chenal, CIRBA, Abidjan, Cote d'Ivoire; Kla Albert Minga, CNTS, Abidjan, Cote d'Ivoire; Emmanuel Bissagnene and Aristophane Tanon, CHU Treichville, Cote d'Ivoire; Moussa Seydi, CHU de Fann, Dakar, Senegal; Akessiwe Akouda Patassi, CHU Sylvanus Olympio, Lomé, Togo. Pediatric cohorts: Sikiratou Adouni Koumakpai-Adeothy, CNHU, Cotonou, Benin; Lorna Awo Renner, Korle Bu Hospital, Accra, Ghana; Sylvie Marie N'Gbeche, ACONDA CePReF, Abidjan, Ivory Coast; Clarisse Amani Bosse, ACONDA_MTCT+, Abidjan, Ivory Coast; Kouadio Kouakou, CIRBA, Abidjan, Cote d'Ivoire; Madeleine Amorissani Folquet, CHU de Cocody, Abidjan, Cote d'Ivoire; François Tanoh Eboua, CHU de Yopougon, Abidjan, Cote d'Ivoire; Fatoumata Dicko Traore, Hopital Gabriel Toure, Bamako, Mali; Elom Takassi, CHU Sylvanus Olympio, Lomé,Togo; Coordinators and data centers: François Dabis, Renaud Becquet, Charlotte Bernard, Shino Chassagne Arikawa, Antoine Jaquet, Karen Malateste, Elodie Rabourdin, Thierry Tiendrebeogo, ADERA, Isped & INSERM U1219, Bordeaux, France. Sophie Desmonde, Julie Jesson, Valeriane Leroy, Inserm 1027, Toulouse, France. Didier Koumavi Ekouevi, Jean-Claude Azani, Patrick Coffie, Abdoulaye Cissé, Guy Gnepa, Apollinaire Horo, Christian Kouadio, Boris Tchounga, PACCI, CHU Treichville, Abidjan, Côte d'Ivoire.

**Author contribution.** C.B., M.S. and N.F.N. designed the study and wrote the protocol. C.B., H.A.L., V.P. and H.F. managed and analyzed the data, and H.A.L. wrote the first draft of the manuscript. B.N. and D.D. realized the inclusion of the patients and collected the data under the supervision of J.M.T., M.S. and N.F.N.. I.N. trained and supervised the team as the referring psychiatrist. S.Z., as an expert in IPT, trained the team and supervised group IPT throughout the project. They helped in the analyses of the data. C.B. and H.F. critically reviewed the early draft of the manuscript. All authors critically reviewed and have approved the final manuscript.

**Financial support.** Supported by the National Institute of Mental Health (NIMH), National Cancer Institute (NCI), the Eunice Kennedy Shriver National Institute of Child Health & Human Development (NICHD) and the National Institute of Allergy and Infectious Diseases (NIAID) of the U.S. National Institutes of Health (NIH), as part of the International Epidemiologic Databases to Evaluate AIDS (IeDEA) under Award Number U01AI069919. The content is solely the responsibility of the authors and does not necessarily represent the official views of the National Institutes of Health.

**Competing interests.** The authors declare that they have no conflicts of interest.

**Ethical standard.** This work complies with the ethical standards of the relevant national and institutional committees on human experimentation and with the Helsinki Declaration of 1975, as revised in 2008.

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
