## [Reviewer Report]

This is a very well done paper.It is clearly written and interesting.My comments are minor. The paper is lengthy and if anything needs to be cut it could be the case material. This depends on the journals ability to accept longer articles.

The Impact statement summarizes what was done but does not say what was found .eg “It identifies requirements..clinics” which is the statement but they might say instead what requirements were identified.,otherwise this is just another summary. This is an issue throught out the impact statement.. What was the impact rather than they had an impact. The authors should make clear up front that this is an implementation presentation and does not present efficacy data.. The fact that such a large percent did well is encouraging but the same number might have done well if they came and talked about the weather. There is no comparison group.5

---

## [Reviewer Report]

Reviewers Comments

Thank you for inviting me to review this manuscript. While it addresses an important topic and provides valuable insights into the implementation of group interpersonal therapy (IPT) in Senegal, several critical areas need to be addressed before it can be considered for publication.

These include clarifying the limitations of IPT from previous studies in Africa, clarifying the quantitative study design, improving the description and validity of study variables and the statistical analysis methods, and providing detailed information on measurement tools and their psychometric properties. With these revisions, the manuscript may have the potential to make a significant contribution to the field of global mental health.

When a new intervention is introduced into a country, it typically needs to be adapted to the local cultural context before implementation. This adaptation process ensures that the intervention aligns with the target population’s social, cultural, linguistic, and systemic realities, which increases its acceptability, feasibility, and effectiveness. Implementing an intervention without considering cultural and contextual factors can lead to poor engagement, misunderstanding, and failure. Authors should discuss the frameworks that guided their adaptation of IPT to suit participants in Senegal.

*Below are major revisions that need to be addressed

1. The study title, “Implementation of group Interpersonal Therapy to treat depression in People Living with HIV: a first evaluation of IPT dissemination in West Africa,” is not appropriate. Senegal is not equal to West Africa. The propagation of “one size fits all” is detrimental to the health of vulnerable populations in Africa.

- Senegal is just one of many countries in West Africa, each with distinct health systems, cultural practices, and population characteristics. Using “West Africa” in the title implies that the study findings represent the whole region, which is not supported by the data.

- The authors should specify “Senegal” instead of “West Africa,” in their title.

2. The Abstract

- Background: This section should state the use of IPT in other African countries and what its effects have been.

- Include the objective of the study

- Method: Revise to include the sample size, & simple description of study participants.

- The phrase “defined based on the literature” is vague and lacks detail on operational definitions or metrics used to measure acceptability and feasibility. Include specific criteria or indicators (e.g., attendance rate, dropout rate, or patient satisfaction) and their thresholds. For example, Acceptability was measured by......... Feasibility was evaluated based on ......

- Specify the confounders and how they were accounted for in the models.

- The description of using “general linear mixed models” to describe changes over time is insufficient. It lacks details about: the type of model (e.g., random intercepts, random slopes). Variables included (e.g., fixed and random effects).

- Specify the process for qualitative data collection, coding, and analysis.

- Key metrics, such as “high attendance (97%)” and “low drop-out (<5%),” are mentioned without providing additional context, such as the total number of participants or absolute numbers. The magnitude of symptom and functioning improvement (e.g., effect sizes) is also not included.

- The term “improvements were maintained 3 months later” is vague and does not clarify whether improvements were statistically significant or clinically meaningful at follow-up.

- While the success of “opposite-sex group facilitation” and “train-the-trainer strategy” is noted, the context or supporting evidence for their success is absent. Include specific metrics or qualitative feedback to illustrate the success of these adaptations.

- Conclusion: One key issue with this conclusion is that it overgeneralizes the success of implementation without sufficiently addressing limitations or challenges identified in the study.

3. Impact Statement

- The statement implies that findings from Senegal can be directly applied to the broader Sub-Saharan Africa region, without considering the significant diversity in healthcare systems, cultural contexts, and resource availability across countries in the region. This assumption risks oversimplifying the complexities of implementing group IPT in other contexts.

- Extrapolating findings beyond the study’s scope without taking into consideration of the evidence from other countries on the limitations of IPT, undermines the scientific rigor and can mislead policymakers or funders.

- Authors should limit conclusions to Senegal and recommend further studies to test the intervention in other countries before proposing regional scale-up.

- The second issue with this impact statement is that: While the statement highlights the success of group IPT and its adaptations, it downplays the significant challenges identified in the study, such as issues with depression screening, diagnosis communication, and confidentiality. These are critical for assessing the intervention’s scalability and sustainability.

- This reviewer suggests that include a balanced discussion of both successes and limitations to provide a realistic roadmap for future implementation.

I recommend that the authors read the World Bank publication on IPT in Uganda and put into consideration these findings as they revise their impact statement.

World Bank. (2022). Therapy, mental health, and human capital accumulation among adolescents in Uganda. Washington, D.C.: World Bank. Retrieved from https://documents1.worldbank.org/curated/en/099552207102441963/pdf/IDU1007db5cd16b2f146811a516124d1708f3085.pdf

While the Senegal study focuses on adults living with HIV, it is crucial to recognize that IPT may interact with population-specific factors (e.g., age, gender, socioeconomic status) differently. This warrants caution when generalizing results across contexts or populations.

The Senegal study only followed participants for 3 months, which is too short to assess the sustainability of IPT’s effects.

The Ugandan study highlights that while IPT may show promising short-term benefits, these effects may diminish over time, especially without ongoing support or systemic changes.

If the long-term efficacy of IPT remains unproven, recommending broader implementation may be premature and could lead to resource misallocation or unmet expectations.

4. Introduction

*The introduction states that “Group IPT was first applied in sub-Saharan Africa (SSA) with positive results in Uganda (Bolton et al. 2003)” and goes on to highlight its effectiveness in treating depression in people living with HIV (PLWH). However, evidence from Uganda and other SSA countries shows mixed results, particularly regarding sustainability and effectiveness in men.Bolton et al 2007 showed the intervention was not effective in males and he recommended that other interventions be developed for males in Africa. This is not acknowledged.

*Overgeneralizing IPT’s success without revealing its limitations (e.g., gender-specific differences in efficacy) can mislead readers about its universal applicability.

*Authors should provide a balanced discussion by acknowledging both positive outcomes and challenges from prior studies, including population-specific limitations (e.g., effectiveness primarily in women).

*Include a brief mention of task-shifting challenges and how the study hopes to address or navigates them.

*The introduction describes PHASE-1 as a success based on “high acceptability, feasibility, and benefits for PLWH,” without providing specifics or acknowledging potential limitations (e.g., short follow-up period, small sample size, or lack of scalability analysis).

5. Methods

*It is clear that this is a mixed methods study. But what is the study design under quantitative methods?

Organizational changes

*While the shift in screening and diagnosis responsibilities and the introduction of supervisory roles qualify as organizational changes, elements like remote training and opposite-sex facilitation are better classified as adaptations to implementation processes rather than structural or organizational changes. The authors should delineate organizational changes from implementation adjustments to avoid confusion.

Study population:

*This section has not described the study population

*I cannot find the study selection criteria for participants. What was the inclusion and exclusion criteria?

*In PHASE-1, a PHQ-9 score of ≥5 was used to refer participants for clinical diagnosis, while in PHASE-2, a higher threshold of ≥10 was applied. This inconsistency introduces a potential bias in participant selection between the two phases, as individuals with mild depressive symptoms (PHQ-9 scores 5–9) were excluded in PHASE-2.

*The authors should justify the change in PHQ-9 cut-off scores and discuss its implications for the comparability of results.

*The sampling criteria for stakeholders, such as facilitators, supervisors, physicians, and heads of services, are vague. Whether these individuals were purposefully selected based on specific characteristics or randomly chosen is unclear.

*Authors should describe how stakeholders were selected (e.g., based on their involvement in the intervention, expertise, or availability).

*While 22 PLWH were purposefully selected for qualitative interviews, the specific sampling criteria (e.g., age, marital status, profession, and levels of suicidal risk) are only briefly mentioned. It is unclear how these factors were operationalized or prioritized.

*Specify how participants were chosen based on these criteria and whether any efforts were made to ensure diversity in the sample (e.g., stratified purposeful sampling).

*Without a description of how suicide risk was assessed, it is unclear whether standardized tools or clinical judgment were used, which raises questions about consistency and validity in identifying high-risk individuals.

*If suicide risk assessment was inconsistent or subjective, it could introduce bias into the selection process, excluding or including participants inappropriately.

*Assessing and managing suicide risk is a critical ethical consideration in mental health studies. Without detailing the process, the authors may fail to demonstrate adherence to ethical guidelines for participant safety.

Quantitative Data Collection Procedures

*The section does not describe how data was collected or who collected it. Specify who collected the data, their training, and whether standardized protocols were used.

*Feasibility indicators are broadly mentioned (e.g., “attendance of therapy sessions,” “risk of suicide”), but specific thresholds or criteria for feasibility are not provided. Define each feasibility indicator with measurable criteria.

*To enhance the clarity and rigor of the study, all variables, including sociodemographic variables, need to be described in detail, specifying how they were measured and incorporated into the regression models (e.g., categorical or continuous). Additionally, the psychometric properties of the measurement tools, such as reliability, validity, and any cultural adaptations, must be reported to ensure their appropriateness for the study population. This information is essential for evaluating the reliability and validity of the findings."

Quantitative data Analysis

*Why are the authors comparing phase 1 and phase 2?

*What objective of the study is this comparison addressing?

*There are several methodological and reporting issues in this description of data analysis.

*The description of the GLMM is vague. It does not specify:

o The dependent variables modeled (e.g., PHQ-9 scores or WHODAS scores).

o Fixed and random effects included (e.g., whether Time and PHASE were modeled as fixed effects, and whether individual participants were modeled as random effects).

o Interaction terms, if any (e.g., Time × PHASE interaction to assess differences in symptom improvement over time across PHASES).

*Explicitly describe the GLMM structure and rationale for variable inclusion.

*The section lists the Wilcoxon test, Chi-square test (Chi-2), and Fisher’s exact test but does not explain which variables were analyzed with each test or why these specific tests were chosen.

*Clearly link each test to the type of data being analyzed (e.g., Wilcoxon for non-normally distributed continuous data, Chi-2 for categorical data).

*The description of the analysis for “opposite-sex facilitation” is unclear. It does not specify:

o The dependent variable (e.g., depressive symptoms, PHQ-9 scores).

o The inclusion of interaction terms (e.g., Time × Opposite-Sex Facilitation) to evaluate differences over time.

o How opposite-sex facilitation was measured or categorized (binary or continuous).

Provide specifics on how opposite-sex facilitation was modelled and interpreted.

*It is unclear whether missing values were handled through imputation, sensitivity analysis, or other methods. Specify how missing data were handled and justify the chosen method.

Results

The validity of the reported results cannot be fully evaluated because the described analysis methods are incomplete and potentially inappropriate. Specifically, the lack of description of measurement tools such as suicide risk assessment tools, the lack of clarity around the structure of the generalized linear mixed models (GLMMs), and handling of missing data raise concerns about the validity of the findings. Without addressing these issues, it is not possible to interpret the results confidently and to comment on the discussion of these results

---

## [Reviewer Report]

Thank you for the opportunity to review this interesting manuscript. I commend the authors for the rigorous study of IPT among PLHIV in Senegal. The manuscript is well written. Main suggestions are related to scope of manuscript. The authors have made some interesting comparisons between the two phases of the study. However, making comparisons and evaluating effectiveness and implementation outcomes seemed to broaden the scope of the paper to the point where it made it difficult to follow. Tightening the scope and the aims of the paper to focus just on Phase 2 implementation and effectiveness outcomes (and then making comparisons int he discussion with phase 1 outcomes) would benefit the readability and highlight the contribution of this study. Additionally, a true mixed methods analysis would triangulate the qual/quant data. This triangulation is missing. Specific points pertaining to manuscript sections are described below.

Abstract/title

Instead of benefits, perhaps “effectiveness” more accurate descriptor

Methods: “Acceptability and feasibility were defined based on the literature.”

Please add more detail here. Frameworks for sustainability and implementation should be included.

Results:

“Adaptations, such as opposite-sex group facilitation and use of train-the-trainer strategy, were

successful.”

Would be helpful to include stats.

Results missing numbers and descriptions of participants.

“Depressive symptoms and functioning improved drastically”

Instead of drastically, use “significantly”

INTRODUCTION:

Implementation Adaptations:

The adaptations of using different sex facilitators and train-the-trainer models are fairly standard and straightforward. While I appreciate the testing of these, the background and rationale for why these adaptations would be important or significant is missing. It is fairly well established that non-specialists can effectively deliver evidence-based interventions and a train-the-trainer model is often used when scaling up programs. More justification for the significance of this focus is needed.

minor- line 94 background - “Expanding group IPT beyond Dakar to these areas could

help reduce this disparity gab”

Change to gap.

AIMS:

Using a mixed-methods approach, this study aimed to: 1) confirm group IPT’s acceptability, feasibility and benefits in these different contexts; 2) assess the impact of organizational changes between PHASES 1 and 2; and 3) determine requirements for sustainability.

Hypotheses are missing.

Please describe how the paper is mixed methods. A true mixed methods approach triangulates quant and qual data. As it reads, the methods look like they were done in parallel to one another.

METHODS

Please include more information describing supervision and how competency was established.

The participant numbers are unclear- please specify.

What was the rationale for including people 20 and over and not 18?

More description is needed regarding the measures. For example, regarding the CSQ-8, which four questions were retained and how scored and analyzed? How were measures adapted for context if at all?

How were interview guides revised based on observations and discussions? please give examples

More info needed on delivery of IPT- duration? length of session? any adaptations/tailoring made to intervention?

More description of measures and items- used before in senegal? translation procedures?

Implementation Frameworks:

Authors state that 3 frameworks used to guide the research. How so? Much more detail needed. How were the 3 frameworks integrated? What was the rationale for each and using the 3 together? Results should also be organized by and follow the frameworks. A cohesive inclusion of them throughout the study (background, methods, results, discussion) is needed if they are to be included.

Analysis/comparative focus:

The comparisons between phases made in the analysis and results section not clear in title or justified in background of paper. Tables show p-values instead of tests mentioned in data analysis section (eg chi-square). I wonder if the paper could be better focused by analyzing data in phase 2 only (hybrid effectiveness-implementation study) rather than comparing across datasets. Bringing in data from phase 1 seems to be confusing. Also, phase 1 occurred during COVID, so it’s hard to make inferences about implementation. A general comparison with phase 1 could be done in the discussion section.

More information needed for the qual data analysis. Recommend following COREQ example. How were data coded, by whom, and interrater reliability achieved?

RESULTS

SAMPLE CHARACTERISTICS

Thoughts about why the positive rate is so much higher in phase 2 (61%) as phase 1 (14%)? Does this have to do with a nonspecialist in MH confirming the diagnosis?

Recommend being specific about percentages in sample characteristics.

Lines 241-249: It seems like this is qual information that’s being presented, but it’s unclear. No quotes are used and thematic organization is not present. If using both qualitative and quantitative data, reorganizing by theme would be helpful and ensuring that the data speak to one another if mixed methods is important.

Heading title “Depressive symptoms evolution and other benefits of group IPT”… suggest “depressive symptom reduction…”

It’s unclear when data from phase 1 is being presented, phase 2, or combined. Example: “An overall improvement was observed during PHASES

256 (β = -1.1, CI95% [-2.1, -0.2], p = 0.015).” Is this combined data?

A focus of the results is screening, but aside from the qual data, it’s unclear how screening and access to it was measured.

To evaluate the train-the-trainer model, some kind of fidelity/competence score would be needed. Please consider if this should be a main focus of the paper.

DISCUSSION:

Highlighting of the key contribution missing in discussion.

Limitations:

“the two PHASES was performed retrospectively, resulting in analyses that were not originally planned. Therefore, the interpretation of the results should be undertaken with caution.”

This retrospective analysis comes through in the presentation of the paper and detracts from the cohesiveness. As recommended above, consider using data from just phase 2.

---

## [Reviewer Report]

This is a very well done paper with interesting details about the implementation of IPT in Senegal. This is a descriptive longitudinal study of the implementation of IPT. It is not a clinical trial but there have been over 150 trials of the treatment. The paper reports on acceptance ,feasibility benefits and issues.The clinical material is useful in making the points clear. Interesting,is the concern about stigma if the term depression is used.This has not been noted in other reports from Africa.

I have only one concern. There is much detail about most aspects of the study and this is useful.But one is missing.Who are the facilitators. What level education do they have and how are they chosen.It is noted that they have to abide by the rules page 39 but who are they.? Also this work comes out of an HIV program.Is it being used in the same way in non HIV programs and what do the authors envision? This is very nice work. The authors may want to note that reference 11 has reports on IPT use in other african countries